# MULTI-TASK IMAGE-TO-IMAGE DIFFUSION MODELS WITH FINE-GRAINED CONTROL

## ABSTRACT

Diffusion models have recently been applied to various image restoration and editing tasks, showing remarkable results in commercial products, e.g., Adobe Photoshop. While recent approaches to text-based editing have shown flexibility and great editing capacity, they still lack fine-grained control and/or multi-task compositing capabilities. In everyday applications, however, having a single tool for image editing with detailed user control across multiple tasks is highly preferred. This paper proposes a multi-task image-to-image diffusion model that allows fine-grained image editing among multiple tasks within a single model. Our approach builds upon conditional diffusion models and jointly models the input images and the input compositing effects, including motion blur, film grain, colorization, image sharpening, and inpainting. We present a novel input conditioning formulation and observe that using explicit binary task activation labels and cross-attention-based feature conditioning are key to allowing the model to achieve multi-task editing. In addition, we introduce a novel benchmark dataset for image compositing effects with standard image metrics for advancing the state of the art. Our approach can manipulate natural images with fine-grained, disentangled user control on single- and multi-task editing setups and generalizes well across different domains and even to unseen data distributions. We present experimental results on various compositing tasks to show that our approach outperforms existing techniques and baselines.

## 1 INTRODUCTION

Diffusion models have recently made significant progress in image generation Ho et al. (2020); Dhariwal & Nichol (2021), attracting large interest to generative AI. The famous stable diffusion approach by Rombach et al. (2022) generates impressive images based on text prompts, revolutionizing the entire art industry. Many commercial photo editors now integrate AI-based editing modules based on stable diffusion, such as Adobe Photoshop, Luminar Neo, Canva, and Lensa.

Diffusion-based models have been applied to various image-to-image translation tasks, including image super-resolution Saharia et al. (2022c), inpainting Saharia et al. (2022a); Lugmayr et al. (2022), denoising Kawar et al. (2022a); Zhu et al. (2023); Murata et al. (2023), deblurring Whang et al. (2022); Ren et al. (2022), stroke-based editing Meng et al. (2021) and text-based editing Brooks et al. (2023); Parmar et al. (2023). Existing work on diffusion-based image editing focuses more on image restoration tasks, *e.g.,* denoising and deblurring. The inverse tasks, *e.g.,* motion blur and film grain synthesis, are yet to be explored.

Among the different types of approaches, text-based editing stands out as one of the hottest topics in diffusion-based image editing since it provides users direct control via intuitive text prompts Brooks et al. (2023); Parmar et al. (2023); Tumanyan et al. (2023). Localized editing can be achieved by integrating masks Avrahami et al. (2022) or feature injection Hertz et al. (2022); Tumanyan et al. (2023). Nevertheless, fine-grained control is not ensured as the text encoders are not trained with this intent but instead for general-purpose tasks. Besides, they are initialized from pretrained models, *e.g.,* BERT Devlin et al. (2018) and CLIP Radford et al. (2021), which are frozen during training.

However, photo editing with fine-grained control is a primal need for end users. Another feature of great demand is multi-task editing, which allows for performing multiple tasks within a single model. For instance, in film post-production, artists need to incrementally add compositing effects, such as motion blur, film grain, and colorization, to the processed frames. Thus, a single model that

performs multiple editing tasks simultaneously is highly desirable for efficiency and preventing the accumulation of undesired artifacts. Unfortunately, multi-task training for image-to-image translation is relatively under-explored Saharia et al. (2022a).

In this paper, we focus on the problem of multi-task image editing, which aims to learn a single model for multiple image-to-image translation tasks. Such a problem is exceptionally challenging as the model needs to learn a shared feature space and disentangle each task simultaneously for multi-task editing capabilities. On top of that, given a set of $N$ editing tasks, there are $\sum_{k=1}^{N} \binom{N}{k}$ target task combinations. As such, providing all possible combinations in the training data turns intractable, especially with increasing editing tasks.

To tackle the above limitations, we propose a multi-task image-to-image diffusion-based framework that is trained on single-task editing data but generalizes to multi-task edits. Specifically, we design a novel multi-modal input conditioning formulation that provides explicit user control over each task. It includes image conditions applied via image concatenation and an editing vector condition applied via cross-attention features. While cross-attention-based conditioning has been proposed before Rombach et al. (2022), it has mainly been applied to text-to-image diffusion models to allow for generalizing to complex text prompts. In our formulation, the editing vector condition represents task-specific editing handlers to provide explicit fine-grained user control over each task. We also add binary task activation labels, represented as one-hot encodings, to the input editing vector. We observe that cross-attention-based conditioning and explicit binary labels are key to allowing the model to generalize on combined tasks.

We train our model on different image editing/compositing tasks, including motion blur synthesis, film grain synthesis, color transfer, inpainting, and sharpening. We remark that motion blur and film grain synthesis are less explored in the literature. However, they are of prime importance for practical use in professional pipelines, *e.g.,* film post-production and VFX, as it is non-trivial to synthesize these effects with fine control and ensure naturally-looking results in multi-task editing setups. Our proposed input conditioning provides explicit task activation and user control, including the level of motion blur, the size of film grain, the reference image for colorization, the amount of sharpness, and the inpainting region. We provide extensive experimental results to show that our method outperforms existing baselines. Our main contributions can be summarized as follows:

- We propose a novel diffusion-based architecture for multi-task image-to-image translation problems, providing users with fine-grained control over each task.

- We propose a novel conditioning formulation that allows the model trained on data with single-task edits to generalize well to multi-task editing. We identify that an explicit binary task activation vector and cross-attention-based conditioning are key to model generalization.

- Our approach yields competitive results on single-task editing and state-of-the-art results for multi-task editing. It also generalizes to novel datasets not viewed at training time.

- We introduce a novel benchmark dataset based on REDS Nah et al. (2019) with standard image metrics for advancing the state of the art.

## 2 RELATED WORK

**Image-to-image translation.** Historically, Generative Adversarial Networks (GANs) Goodfellow et al. (2014); Radford et al. (2015); Creswell et al. (2018); Brock et al. (2018) have been a popular architectural choice for image-to-image translation tasks. Image-to-image translation tasks have been attempted by conditional generative models Isola et al. (2017); Wang et al. (2018); Kim et al. (2020) with success across a wide variety of domains, such as semantic labels to photo-real image or colorization. However, this requires a separate paired dataset per task and a separate model for each task. Other methods apply edits within the GAN latent space Shen et al. (2020); Kim et al. (2021) using vector arithmetic. This can yield interesting results due to the disentangled nature of the latent space of a well-trained GAN. However, it does not allow fine-grained control of attributes and is prone to artifacts. GAN inversion Xia et al. (2021) has also been used to create paired data for the conditional generative paradigm when this is not feasible to obtain Wu et al. (2022). This suffers from similar drawbacks, namely artifacts in the output and the limitation of the abilities of the encoder to encode the source image into the latent space. GANs have additionally been used for style transfer Taigman et al. (2017); Liu et al. (2019), for example with CycleGANs Zhu et al.

(2017), where a cycle consistency loss is enforced to map between a source image domain and a target domain in an unsupervised manner. This overcomes the limitation of requiring paired data, but it does not allow for fine-grained control of the image modification. Compositing effects, such as motion blur Luo et al. (2018) and film grain addition or removal Ameur et al. (2023), are less explored and we seek to remedy this in our proposed method.

**Diffusion techniques.** Diffusion models Sohl-Dickstein et al. (2015) have received considerable attention since their initial conception and deployment for image generation Ho et al. (2020); Nichol & Dhariwal (2021); Ho et al. (2022); Song et al. (2022a); Rombach et al. (2022) as they produce very high-quality samples, are more stable during training and capable of more diverse image generation than GANs. They have shown good performance on several image restoration tasks Kadkhodaie & Simoncelli (2021); Sasaki et al. (2021); Kawar et al. (2022a); Saharia et al. (2022c); Zhu et al. (2023); Murata et al. (2023), although multi-task editing is less explored. For multi-task editing without re-training the diffusion model for each task, guidance from an external classifier or regression model can be added at sampling time Wolleb et al. (2022); Ho & Salimans (2022); Kawar et al. (2022b).

Text-based editing Ramesh et al. (2022); Saharia et al. (2022b); Rombach et al. (2022) seeks to control the structure of a generated image, which can be achieved by using spatial attention maps generated from a reference prompt to edit part of the generated image Hertz et al. (2022); Tumanyan et al. (2023). While designed for synthetic images, real images can be edited through inversion. However, this does not always produce an accurate result Zhang et al. (2023). Further research uses inversion to create paired training data for a conditional model Brooks et al. (2023). While this allows generalization to real images, it does not support fine-grained control. For more advanced control of text-to-image models, *e.g.,* with segmentation or edge maps, a hypernetwork-adjacent architecture can be used Zhang & Agrawala (2023). However, paired training data is still required, and multi-task conditioning was not investigated. An end-to-end approach, which may be more desirable, has been accomplished with adapted text-to-image models that use latent embeddings of an object created with a vision transformer Song et al. (2022b). Although this retains the object's semantics and holistically matches its color, lighting, etc. in the scene, it does not retain the fine detail of the image and gives little control to the user.

**Multi-task image editing.** As training a model for each domain would be intractable when building large-scale pipelines, the "multi-task" approach capable of editing within many domains has great value within image-to-image editing Yu et al. (2018); Qian et al. (2022). Some approaches seek to accomplish this with a GAN-based framework through additional class labels Choi et al. (2018) or mapping networks Choi et al. (2020). This adds computational overhead and complexity to the training. It also adds additional modes to the network; GANs are known to drop modes or collapse entirely to only one mode Zhang et al. (2018b). To alleviate these issues, other work Saharia et al. (2022a) uses diffusion models that more completely represent all the modes found in the data distribution. Palette introduces a diffusion model trained on a dataset exhibiting various image degradations with the goal of a generalist model capable of multiple tasks (colorization, inpainting, uncropping, and JPEG restoration). Although it shows the success of training diffusion models on multiple tasks simultaneously, it does not fully specify the task at training time, and therefore, there is a lack of control over the type and amount of the image restoration. Our approach builds on this work by introducing additional conditioning to the model for more fine-grained control of the output.

## 3 METHOD

Our task is to train a single diffusion model on multiple image-to-image translation tasks simultaneously, with fine-grained user control over each individual task.

### 3.1 IMAGE-TO-IMAGE DIFFUSION MODELS

First, we briefly overview our baseline framework derived from Palette Saharia et al. (2022a), which is further built upon conditional diffusion models for image-to-image translation. Please refer to the supplementary document for a general introduction to diffusion models.

Conditional diffusion models can be expressed as a sequence of denoising autoencoders $\epsilon_\theta(\mathbf{x}, \mathbf{y}_t, t)$, where $\mathbf{y}_t$ is the noisy version of $\mathbf{y}$ at the diffusion time step $t$. The conditioning of source image $\mathbf{x}$ is applied by concatenating $\mathbf{y}_t$ with $\mathbf{x}$. Palette's loss is defined as:

$$\mathcal{L}_{Pal} = \mathbb{E}_{\mathbf{x}, \, \mathbf{y}, \, \epsilon \sim \mathcal{N}(0,1), \, t} \left[ \left\| \epsilon_\theta \left( \mathbf{x}, \mathbf{y}_t, t \right) - \epsilon \right\|_2^2 \right], \tag{1}$$

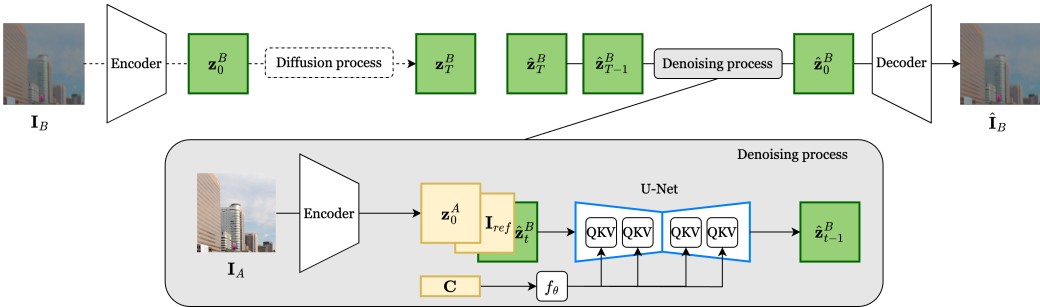

Figure 1: **Architecture of the proposed multi-task image-to-image diffusion model.** Given a source image $\mathbf{I}_A$, the denoising U-Net uses the corresponding latent code $\mathbf{z}_0^A$ as guidance and constructs the target image $\hat{\mathbf{I}}_B$ using multi-modal input conditions: the reference image $\mathbf{I}_{ref}$ and the editing vector $\mathbf{C}$. *Top:* Diffusion and denoising processes. *Bottom:* Zoom-in architecture for a single denoising step.

where $\mathbf{y}_t = \sqrt{\overline{\alpha}_t}\,\mathbf{y} + \sqrt{1-\overline{\alpha}_t}\,\epsilon$, with $\overline{\alpha}_t$ being the noise level indicator at time step $t$.

### 3.2 MULTI-TASK IMAGE-TO-IMAGE DIFFUSION MODELS

Multi-task learning has been briefly discussed in Palette Saharia et al. (2022a). The multi-task Palette model is trained simultaneously on four tasks: colorization, in-painting, uncropping, and JPEG restoration. This multi-task framework shares the same architecture of the task-specific one, conditioned directly on the input image.

Although the multi-task Palette model achieves comparable results to task-specific models, it provides no user control over each task. Furthermore, the multi-task Palette model learns to map four different domains into the same domain of natural images, akin to an N-to-1 mapping. Note that their current conditioning design does not allow N-to-N mapping either.

To address these concerns, we design a multi-task image-to-image diffusion model that allows explicit fine-grained control over each task. We train the model simultaneously on five image-to-image translation tasks commonly used in film production pipelines: motion blur synthesis, film grain synthesis, color transfer, inpainting, and sharpening. We realize these tasks within a single model, i.e., our model is trained to learn an N-to-N mapping.

**Architecture.** The architecture of our proposed multi-task image-to-image diffusion model is detailed in Fig. 1. We extend Palette model to the state-of-the-art Latent Diffusion Models (LDMs) Rombach et al. (2022), where the diffusion models are trained in the latent space of a pretrained perceptual compression model. As shown in Fig. 1, the source image is denoted by $\mathbf{I}_A$, and the target image is denoted by $\mathbf{I}_B$. We use $VQ$ to denote the pretrained autoencoder. The latent codes corresponding to $\mathbf{I}_A$ and $\mathbf{I}_B$ are denoted by $\mathbf{z}_0^A$ and $\mathbf{z}_0^B$, respectively. Following LDMs, we train the diffusion models in the latent space of $VQ$. Hence, the objective is to learn the target latent code $\mathbf{z}_0^B$ from the source latent code $\mathbf{z}_0^A$ using the color reference image $\mathbf{I}_{ref}$ and the editing vector $\mathbf{C}$. The loss function is expressed as follows:

$$\mathcal{L} = \mathbb{E}_{\mathbf{z}^A,\,\mathbf{z}^B,\,\mathbf{I}_{ref},\,\mathbf{C},\,\epsilon\sim\mathcal{N}(0,I),\,t}\left[\left\|\epsilon_\theta\left(\mathbf{z}_0^A,\,\mathbf{I}_{ref},\,\mathbf{C},\,\mathbf{z}_t^B,\,t\right) - \epsilon\right\|_p^p\right], \quad (2)$$

where $p = 1$. Although $\ell_2$ norm favors sample diversity Saharia et al. (2022a), our experimental results show that $\ell_1$ norm attains better reconstruction quality.

**Conditioning.** One of the core differences between our architecture and Palette's is the input conditioning. Palette is conditioned solely on the source image. To provide better control for end users, we add additional input conditions for each task: blur degree $\beta$ for motion blur synthesis, grain size $\sigma$ for film grain synthesis, color reference image $\mathbf{I}_{ref}$ for color transfer, inpainting mask $\mathbf{M}$ for inpainting, and sharpening amount $\lambda$ and threshold $\mu$ for the sharpening task. In addition, we include the binary task activation labels, i.e., one-hot encodings, to indicate whether the corresponding task is active. The

Table 1: **Input conditioning with fine-grained user control.**

| Via Concatenation | Via Cross attention |
|---|---|
| Input Image: $\mathbf{I}_A$
Color Reference: $\mathbf{I}_{ref}$ | blur degree: $\beta \in [0,1]$
grain size: $\sigma \in [0,0.1]$
sharp. amount: $\lambda \in [1,3]$
sharp. threshold: $\mu \in [0,0.1]$
bin. task labels $[e_1,e_2,e_3,e_4,e_5]$ |

scaling factors $[\beta, \sigma, \lambda, \mu]$ and the binary task activation vector $[e_1, e_2, e_3, e_4, e_5]$ are concatenated into an editing vector $\mathbf{C} = [\beta, \sigma, \lambda, \mu, e_1, e_2, e_3, e_4, e_5]$.

Our proposed model is conditioned on the source latent code $\mathbf{z}_0^A$, the color reference image $\mathbf{I}_{ref}$, and the editing vector $\mathbf{C}$. As such, our model employs multi-modal input conditioning. Note that the conditioning of the source latent code $\mathbf{z}_0^A$ and the color reference $\mathbf{I}_{ref}$ is applied via concatenation. For the color reference conditioning, we use the RGB image $\mathbf{I}_{ref}$ down-sampled to the spatial size of $\mathbf{z}_0^A$, since we find that RGB input conditioning yields more accurate color transfer than compressed conditioning. In view of the high-quality results achieved by recent text-to-image diffusion models, we apply the editing vector $\mathbf{C}$ using a cross-attention conditioning mechanism, as proposed by Rombach et al. (2022). Tab. 1 gives a clear summary of our conditioning.

The editing vector $\mathbf{C}$ is passed through a mapping network $f_\theta$ to get the features, which are in turn passed into the cross-attention layers in the denoising U-Net. The mapping network $f_\theta$ is realized as two linear layers with an in-between sigmoid linear unit activation. The $VQ$ autoencoder and the denoising U-Net are initialized from a pretrained text-to-image model Rombach et al. (2022).

## 4 RESULTS

### 4.1 IMPLEMENTATION DETAILS

**Datasets.** We build upon REalistic and Dynamic Scenes (REDS) Nah et al. (2019) to create a custom dataset for training. REDS contains 300 videos ($1280 \times 720$ resolution), collected for video deblurring and super-resolution tasks. We use a {240-30-30} split for training, validation, and testing, respectively. Each video sequence contains 100 pairs of sharp and blurry frames. To process our custom dataset, we crop a $256 \times 256$ patch randomly from each pair of frames. As a result, we obtain cropped pairs of {sharp, motion blur} images.

We utilize off-the-shelf algorithms to synthesize editing effects for the other tasks. We run the method proposed by Newson et al. (2017) for film grain synthesis. Following Ameur et al. (2022), we generate grain effects for each sharp image at five different radii $\{0.010, 0.025, 0.050, 0.075, 0.100\}$. We adopt the algorithm proposed by Reinhard et al. (2001) for global color transfer. The color reference image is another sharp image randomly selected from the training set. For the inpainting task, we utilize the algorithm proposed by Yu et al. (2019) to generate free-form masks for each sharp image. Finally, for the sharpening task, we apply the unsharp masking formula Polesel et al. (2000) given a user-defined sharpness intensity and threshold. Please refer to the supplementary document for further implementation details on the generation of our synthetic editing effects.

**Training.** Our model is trained on a single NVIDIA A10 GPU with 24GB RAM, using a batch size of 12 for $45K$ train steps. At each training step, each sharp image in the batch is randomly labeled with a binary vector "[w/ blur, w/ grain, w/ color transfer, w/ inpainting, w/ sharpening]", indicating if a given task is on or off. The label vector is randomly drawn from $\{[0, 0, 0, 0, 0], [1, 0, 0, 0, 0], [0, 1, 0, 0, 0], [0, 0, 1, 0, 0], [0, 0, 0, 1, 0], [0, 0, 0, 0, 1]\}$. Note that the input conditioning is set to zero for each inactive task. For instance, if the color transfer task is off, the color reference image $\mathbf{I}_{ref}$ will be set to zero. If the inpainting task is on, $\mathbf{I}_A$ will be modified by multiplying the free-form mask $\mathbf{M}$.

### 4.2 QUANTITATIVE EVALUATION

**Evaluation metrics.** We evaluate our approach and baseline methods on image editing using four image metrics: $\ell_1$ distance, Structural SIMilarity (SSIM) Wang et al. (2004), Learned Perceptual Image Patch Similarity (LPIPS) Zhang et al. (2018a) and Kernel Inception Distance (KID) Binkowski et al. (2018). We run both pixel-level and perceptual metrics in our experiments for a fair comparison.

We evaluate the proposed multi-task image-to-image diffusion models on the test set of our custom dataset. We compare our method with a baseline framework derived from Palette[1] Saharia et al. (2022a) and a GAN-based method pix2pix Isola et al. (2017). For a fair comparison, we implement Palette++ framework in the latent space of the same pretrained autoencoder, based on the latent diffusion model Rombach et al. (2022), and provide the model with all the additional proposed input conditioning. The input conditioning is a concatenated tensor containing the latent code of the

---

[1] We refer to this modified version as Palette++.

Table 2: **Quantitative evaluation on single-task image editing.** Results on our test dataset for each task: Blur, grain, color transfer, inpainting, and sharpening. Blue denotes our proposed model.

| MODEL | $\ell_1$ ERR.↓ | SSIM↑ | LPIPS↓ | KID↓ | $\ell_1$ ERR.↓ | SSIM↑ | LPIPS↓ | KID↓ |
|---|---|---|---|---|---|---|---|---|
| | MOTION BLUR | | | | FILM GRAIN | | | |
| PIX2PIX | **0.052** | **0.898** | 0.202 | 7.239 | 0.088 | 0.843 | 0.248 | 17.502 |
| PALETTE++ | 0.062 | 0.880 | 0.198 | 1.855 | 0.058 | 0.886 | 0.104 | 1.728 |
| OURS - SINGLE | 0.059 | 0.885 | **0.194** | **1.544** | **0.057** | **0.887** | **0.100** | **1.464** |
| OURS - MULT. | 0.061 | 0.881 | 0.199 | 1.854 | 0.058 | 0.886 | 0.104 | 1.765 |
| | COLOR TRANSFER | | | | INPAINTING | | | |
| PIX2PIX | 0.236 | 0.647 | 0.426 | 54.206 | 0.080 | **0.915** | 0.261 | 11.312 |
| PALETTE++ | 0.074 | 0.908 | 0.094 | **1.369** | 0.075 | 0.860 | 0.128 | 3.203 |
| OURS - SINGLE | **0.064** | **0.922** | **0.083** | 1.390 | **0.073** | 0.869 | **0.119** | **1.856** |
| OURS - MULT. | 0.074 | 0.908 | 0.093 | 1.385 | 0.076 | 0.857 | 0.130 | 3.391 |
| | SHARPENING | | | | | | | |
| PIX2PIX | 0.143 | 0.789 | 0.461 | 29.630 | | | | |
| PALETTE++ | 0.085 | **0.921** | 0.104 | 1.789 | | | | |
| OURS - SINGLE | **0.084** | **0.921** | **0.102** | 1.807 | | | | |
| OURS - MULT. | **0.084** | **0.921** | 0.104 | **1.773** | | | | |

Table 3: **Quantitative evaluation on multi-task image editing.** Results on our test dataset for combined tasks: Blur + grain and blur + grain + color transfer. The multi-task editing results of our model trained on single edits (denoted by our - single) are generated by sequentially applying each model. Blue denotes our proposed model.

| MODEL | $\ell_1$ ERR.↓ | SSIM↑ | LPIPS↓ | KID↓ | $\ell_1$ ERR.↓ | SSIM↑ | LPIPS↓ | KID↓ |
|---|---|---|---|---|---|---|---|---|
| | BLUR + FILM GRAIN | | | | BLUR + FILM GRAIN + COLOR TRAN. | | | |
| PIX2PIX | 0.103 | 0.801 | 0.354 | 16.775 | 0.541 | 0.129 | 0.754 | 30.458 |
| PALETTE++ | 0.074 | 0.803 | 0.309 | 12.257 | 0.194 | 0.608 | 0.430 | 14.214 |
| OUR - SINGLE | 0.088 | 0.763 | 0.261 | 10.578 | 0.160 | 0.564 | 0.463 | 38.823 |
| OURS - MULT. | **0.073** | **0.815** | **0.246** | **6.829** | **0.142** | **0.697** | **0.373** | **12.421** |
| | FILM GRAIN + COLOR TRAN. | | | | GRAIN + COLOR TRAN.+ SHARPEN | | | |
| PIX2PIX | 0.438 | 0.303 | 0.598 | 50.473 | 0.562 | 0.111 | 0.855 | 37.482 |
| PALETTE++ | 0.134 | 0.753 | 0.293 | 8.261 | 0.206 | 0.642 | 0.355 | 8.992 |
| OUR - SINGLE | 0.111 | 0.737 | 0.262 | 15.494 | 0.188 | 0.585 | 0.455 | 54.575 |
| OURS - MULT. | **0.107** | **0.799** | **0.232** | **5.149** | **0.151** | **0.742** | **0.315** | **6.695** |

input image, the color reference image downsampled to the latent spatial size, and the editing vector expanded to the latent spatial size. As for pix2pix, we adapt the input conditioning to include all the necessary information. The input conditioning is a concatenated tensor containing the input image, the color reference image, and the editing vector expanded to the same input spatial size. We train all three methods on the same training set for the same number of epochs.

**Single-task image-to-image translation.** Tab. 2 reports quantitative evaluation results on each image editing task. We add our model trained on single-task data as an additional baseline for comparison. The model is denoted by "Ours - single" in Tab. 2, where a separate model is trained for each single task and applied independently at inference. Pix2pix performs well on motion blur synthesis but not the other tasks, indicating that the model might be overfitting to a specific task during multi-task training. Our final model trained on multi-task edits performs comparable to Palette++ and "Ours - single". Thus, our proposed multi-task approach preserves accuracy on single tasks.

**Multi-task image-to-image translation.** Tab. 3 reports quantitative evaluation results on multiple image editing tasks. For our model trained on single-task data, we apply each task-specific model sequentially to obtain multi-task edits. Our method outperforms the other three approaches by a large margin. This demonstrates that inserting the conditioning via cross-attention layers promotes disentanglement and helps handle multi-task editing. Note that our model generalizes well to multi-task editing despite only being trained on single tasks.

## 4.3 QUALITATIVE EVALUATION

**State-of-the-art Comparison.** We compare visually our method with those of Palette++ and pix2pix. Fig. 2 presents image editing results on single tasks, while Fig. 3 shows multi-task image editing results. Pix2pix yields noticeable artifacts under the multi-task training setup. Our method yields comparable or better results for single-task image editing than Palette++. For multi-task image editing, our tasks outperform Palette++, our model trained on single edits, and especially pix2pix, by

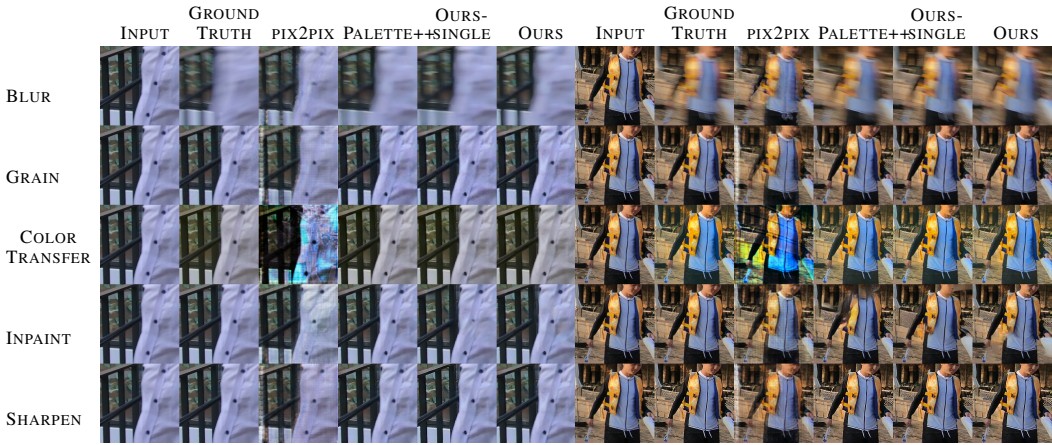

Figure 2: **Qualitative comparison on single-task image editing.**

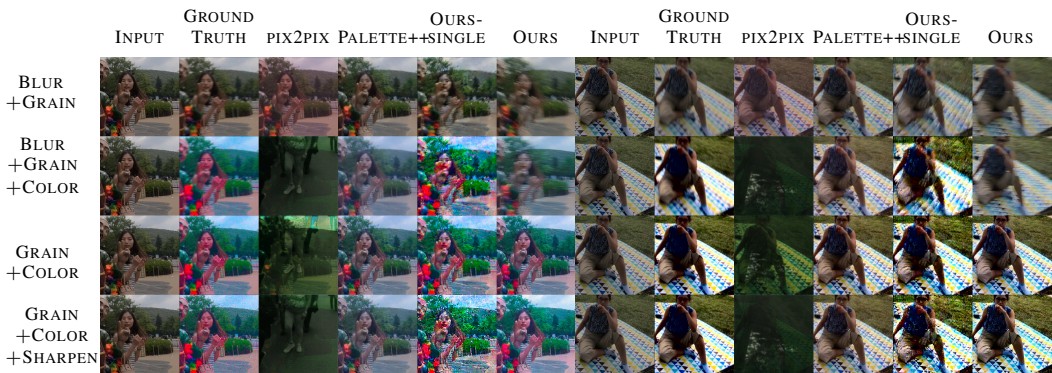

Figure 3: **Qualitative comparison on multi-task image editing.** We evaluate the following combined tasks: blur + grain, blur + grain + color transfer, grain + color transfer, and filmgrain + color transfer + sharpening.

a large margin. Fig. 3 shows that our method succeeds in transferring color when three tasks, *e.g.,* blur, grain, and color transfer, are combined altogether. Overall, our method matches the ground truth better.

**Fine-grained user editing control.** Fig. 4 presents fined-grained progressive editing results for single- and multi-task editing. Editing results were attained by progressively increasing the intensity level of the target effect (motion blur, film grain, sharpening) or randomly switching the reference image (color transfer). The bottom row shows that our model can provide fine-grained control in the multi-task setup as well. Please refer to the supplementary document for more comprehensive experiments on single- and multi-task user editing control.

**Generalization to unseen datasets.** Our method also has good generalization capabilities to unseen datasets. Fig. 5 presents single- and multi-task editing results on FFHQ Karras et al. (2019) and Stanford Cars Krause et al. (2013) datasets, generated by the model trained on our custom dataset. Our model yields plausible editing results for single-task editing, *e.g.,* blur, grain, and sharpening. We also show results on multi-task editing: blur and grain, color transfer and grain, and color transfer, grain, and sharpening. As can be observed from Fig. 5, the same blur degree is given to generate the results in column 2 and 4. Our results show more consistent blur strength compared to those of Palette++ baseline. For the last three columns, the same reference image is given for the color transfer task. Our results show more consistent colorization than Palette++.

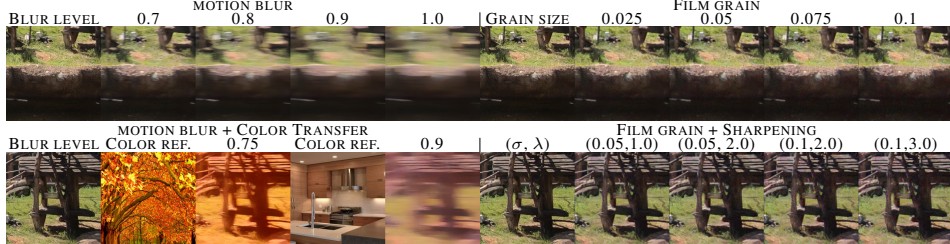

Figure 4: **Fine-grained user control on single- and multi-task editing.** Editing results were attained by progressively increasing the intensity level of the target effect (motion blur and film grain), applying various arbitrary inpainting mask regions (inpainting), or randomly switching the reference image (color transfer).

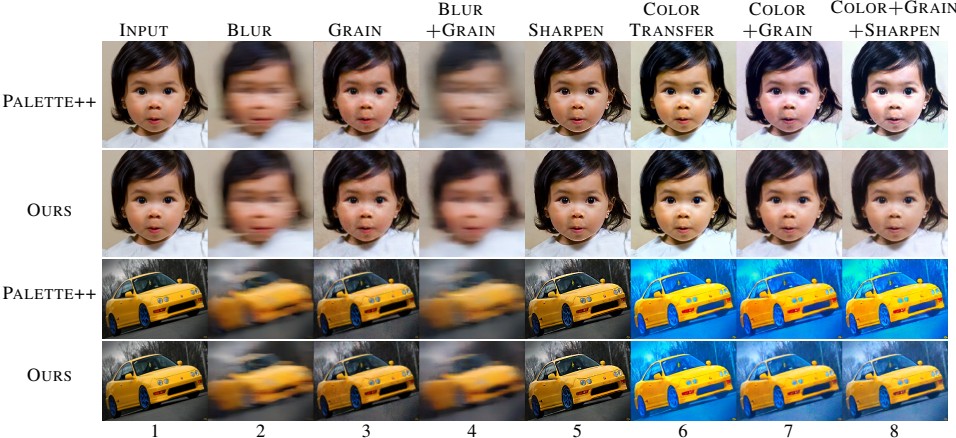

Figure 5: **Multi-task editing results on unseen dataset.** We show single- and multi-task editing results on FFHQ Karras et al. (2019) and Stanford Cars Krause et al. (2013) using the model trained on our custom dataset.

## 4.4 ABLATIVE ANALYSIS

**Binary task activation labels.** The editing vector $\mathbf{C}$ consists of the scaling factors $[\beta, \sigma, \lambda, \mu]$ and the binary task labels corresponding to each task's activation status. Intuitively, these labels might be redundant since they are implicit in the input vector, *e.g.,* blur degree $\beta = 0$ indicates that the motion blur task is disabled. We perform an ablation study to understand their influence on the results, as illustrated in Tab. 4. The binary task activation labels (our approach) slightly improve single-task editing and, by promoting task disentanglement, boost overall performance for multi-task editing.

**Depth of $f_\theta$.** We remark that our architecture's mapping network $f_\theta$ contains two linear layers. Tab. 4 shows quantitatively that this is an optimal choice for the number of layers. We observe that for the case of editing one single task, using a deeper $f_\theta$ yields slightly better results. As for multi-task editing, a performance drop is observed when increasing layers. Hence, we utilize two layers as a balanced trade-off between single- and multi-task editing.

## 5 DISCUSSION

**Limitations.** Although our approach yields good editing results on multi-task cases, task disentanglement is not fully achieved. For instance, Fig. 5 shows color transfer results on the three far-right columns. The results on the last two columns (multi-task editing setup) show that the hue intensity slightly differs from that of column 6 (single-task editing setup). This means that the color transfer task is still somewhat entangled with the graining and sharpening tasks. In addition, we notice that our model fails in the inpainting task when performing multi-task editing. The reason may be that all the other four tasks are global editing effects, except inpainting being local editing. Tab. 3 also confirms that multi-task editing obtains higher error metrics than that of single-task editing. Our approach also lacks pixel-wise crispness inherited from the sampling process of diffusion models. At

Table 4: **Ablation study for architecture choices.** Blue denotes our proposed model.

| MODEL | $\ell_1$ ERR.↓ | SSIM↑ | LPIPS↓ | KID↓ | $\ell_1$ ERR.↓ | SSIM↑ | LPIPS↓ | KID↓ |
|---|---|---|---|---|---|---|---|---|
| | | BLUR | | | | BLUR + GRAIN + COLOR | | |
| W/O BINARY LABELS | 0.061 | 0.881 | 0.201 | 2.014 | 0.240 | 0.581 | 0.443 | 17.639 |
| W/ BINARY LABELS | 0.061 | 0.881 | 0.199 | **1.854** | **0.142** | **0.697** | **0.373** | **12.421** |
| | | BLUR | | | | BLUR + GRAIN + COLOR | | |
| $f_\theta - 0$ LAYER | 0.061 | 0.880 | 0.201 | 2.078 | 0.198 | 0.621 | 0.374 | 14.620 |
| $f_\theta - 1$ LAYER | 0.061 | 0.881 | 0.199 | 1.957 | 0.165 | 0.657 | **0.369** | **12.046** |
| $f_\theta - 2$ LAYERS | 0.061 | 0.881 | 0.199 | 1.854 | **0.142** | **0.697** | 0.373 | 12.421 |
| $f_\theta - 4$ LAYERS | 0.061 | 0.882 | 0.198 | **1.818** | 0.152 | 0.650 | 0.413 | 20.718 |

Table 5: **Quantitative evaluation on contrastive learning (N-pair loss).** Blue denotes our proposed model. Config A and B refer to the contrastive N-pair loss applied to bottleneck features and label features, respectively.

| MODEL | $\ell_1$ ERR.↓ | SSIM↑ | LPIPS↓ | KID↓ | $\ell_1$ ERR.↓ | SSIM↑ | LPIPS↓ | KID↓ |
|---|---|---|---|---|---|---|---|---|
| | | BLUR | | | | FILM GRAIN | | |
| + N-PAIR - CONFIG A | 0.062 | 0.877 | 0.205 | 2.348 | 0.058 | 0.885 | 0.107 | 2.236 |
| + N-PAIR - CONFIG B | 0.062 | 0.880 | 0.200 | 1.923 | **0.057** | **0.891** | 0.120 | 1.851 |
| OURS - MULT. | **0.061** | **0.881** | **0.199** | **1.854** | 0.058 | 0.886 | **0.104** | **1.765** |
| | | BLUR + FILM GRAIN | | | | BLUR + FILM GRAIN + COLOR TRAN. | | |
| + N-PAIR - CONFIG A | **0.073** | 0.798 | 0.277 | 10.662 | 0.213 | 0.592 | 0.394 | 12.607 |
| + N-PAIR - CONFIG B | 0.088 | 0.738 | 0.366 | 43.615 | 0.162 | 0.651 | 0.391 | 12.480 |
| OURS - MULT. | **0.073** | **0.815** | **0.246** | **6.829** | **0.142** | **0.697** | **0.373** | **12.421** |

each sampling step, slight errors are accumulated, resulting in inaccuracies in the synthesized output. Alternative image editing methods have been proposed for inverting existing high-quality images to the latent space of diffusion models Dhariwal & Nichol (2021); Song et al. (2022a); Mokady et al. (2023). We have tried DDIM inversion Dhariwal & Nichol (2021) to obtain the initial noise features from an input image. However, we observe that the obtained noise features are more difficult to edit than random initialized noise features, thus resulting in worse image quality. Please refer to the supplementary for qualitative results.

**Disentanglement.** We have explored different constraints to enhance feature disentanglement for multi-task editing. We have tried different contrastive learning approaches. In particular, we adopted the N-pair loss used in Aksoy et al. (2018), where the loss function is defined as:

$$\mathcal{L}_{N-pair} = \frac{1}{|\mathcal{P}|} \sum_{p,q \in \mathcal{P}} \mathbb{I}[l_p = l_q] \log\left((1 + \exp\left(\|f_p - f_q\|\right))/2\right) \\ + \mathbb{I}[l_p \neq l_q] \log\left(1 + \exp\left(-\|f_p - f_q\|\right)/2\right). \tag{3}$$

This N-pair loss is applied within each batch, where $\mathcal{P}$ denotes the batch of samples, $l_p$ refers to the binary task activation vector of sample $p$, and $f_p$ are the corresponding features. Here, $\mathbb{I}[\cdot]$ returns 1 if the condition is true; 0 otherwise.

We added the N-pair loss to the training objective in our experiments to enhance disentanglement. As for the choice of $f_p$, we have tested both the bottleneck features of the denoising U-Net and the label features $f_\theta(\mathbf{C})$. However, the results in Tab. 5 show that there is no noticeable improvements to the baseline on single-task edits, and even a drop in performance on multi-task edits. Please refer to the supplementary for further analysis.

## 6 CONCLUSION

We demonstrated that multi-task diffusion models can be effective in learning image representations for various photo filter effects, such as motion blur, film graining, sharpening, colorization, and image inpainting. In particular, it is shown that the learned representation enables novel domain generalization towards unseen data distributions and more fine-grained control over generated images, even when it is trained on individual tasks. We achieve this by introducing the binary task activation vector and the cross-attention operations across given input tasks into the proposed conditional diffusion probabilistic formulation. Also, our formulation leads to improved performance in extensive evaluations and comparisons against the state-of-the-art multi-task image editing methods. Task-level feature disentanglements still remain challenging and an interesting research direction. We believe that there will be more advanced solutions for improved task disentanglement, such as task batch scheduling and/or sophisticated contrastive losses, in the future.

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
