# MULTI-TASK IMAGE-TO-IMAGE DIFFUSION MODELS WITH FINE-GRAINED CONTROL: SUPPLEMENTARY MATERIAL

## A  DENOISING DIFFUSION PROBABILISTIC MODELS

Denoising Diffusion Probabilistic Models (DDPMs) Ho et al. (2020) is a class of latent variable models designed to learn a joint data distribution. The forward diffusion process defines a Markov chain that gradually adds Gaussian noise to an image $\mathbf{y}_0$ and produces a set of latent variables $\mathbf{y}_1,...,\mathbf{y}_T$ in the same sample space as $\mathbf{y}_0$. The reverse process is also Markovian, which progressively denoises the input white noise into an image.

DDPMs can be expressed as a sequence of denoising autoencoders $\epsilon_\theta(\mathbf{y}_t, t)$, which are trained to predict the noise added at timestep $t$ from the noisy image $\mathbf{y}_t$. The objective can be defined as:

$$\mathcal{L}_{DM} = \mathbb{E}_{\mathbf{y},\epsilon\sim\mathcal{N}(0,1),t} \left[ \left\| \epsilon_\theta\left(\mathbf{y}_t,\, t\right) - \epsilon \right\|_2^2 \right],  \tag{1}$$

where $\mathbf{y}_t = \sqrt{\overline{\alpha}_t}\,\mathbf{y}_0 + \sqrt{1-\overline{\alpha}_t}\,\epsilon$, with $\overline{\alpha}_t$ being the noise level indicator at time step $t$ and $\epsilon$ being the actual noise added.

## B  ADDITIONAL IMPLEMENTATION DETAILS

In this section, we provide additional details on how to prepare our custom benchmark dataset based on REDS Nah et al. (2019). As presented in the main paper, REDS contains 300 videos ($1280 \times 720$ resolution), with each video sequence containing 100 pairs of sharp and blurry frames. Our custom dataset contains sharp images and five different editing effects applied to each sharp image, including motion blur, film grain, color transfer, inpainting, and sharpening.

**Motion Blur.**  The motion blur effects are directly obtained from REDS dataset, by directly cropping a $256 \times 256$ patch randomly from each pair of frames. The rest of the editing effects are generated using these preprocessed cropped sharp images.

**Film Grain.**  We utilize the open source code in Newson (2017) for the method proposed by Newson et al. (2017) for film grain synthesis. Following Ameur et al. (2022), we generate grain effects for each sharp image at five different radii $\{0.010, 0.025, 0.050, 0.075, 0.100\}$.

**Color Transfer.**  The color transfer effects are generated using the algorithm proposed by Reinhard et al. (2001). For each sharp image, we randomly selected another sharp image from the training set as the color reference image. Specifically, we utilize the open source implementation in Rosebrock (2014).

**Inpainting.**  For the inpainting task, we utilize the algorithm proposed by Yu et al. (2019) to generate free-form masks for each sharp image. Specifically, we adopt the open source implementation in Nippert (2022) using the default configurations.

**Sharpening.**  For the sharpening task, we apply the unsharp masking algorithm proposed in Polesel et al. (2000). This algorithm takes as input a user-defined sharpness intensity and threshold parameter. The pseudocode is presented in Algorithm 1. For the sake of simplicity, we generate sharpening effects using a default Gaussian blur kernel with size $\mu = 3$ and standard deviation $\sigma = 1$.

---

**Algorithm 1** Unsharp Masking

---

1: **Input**: source image $\mathbf{x}$, amount $\lambda$, threshold $\mu$, Gaussian kernel size $s$, Gaussian kernel standard deviation $\sigma$
2: **function** UNSHARPMASK($\mathbf{x}, \mu, \lambda$)
3:     $\mathbf{x}_{\text{blurred}} \leftarrow \text{GaussianBlur}(\mathbf{x}, s = 3, \sigma = 1)$
4:     $\mathbf{M} \leftarrow |\mathbf{x} - \mathbf{x}_{\text{blurred}}| \leq \mu$
5:     $\mathbf{x}_{\text{sharpened}} \leftarrow \mathbf{x} + \lambda \cdot (\mathbf{x} - \mathbf{x}_{\text{blurred}})$
6:     $\mathbf{x} \leftarrow \text{Where}(\mathbf{M}, \mathbf{x}_{\text{sharpened}})$
7:     **return** $\mathbf{x}$
8: **end function**

---

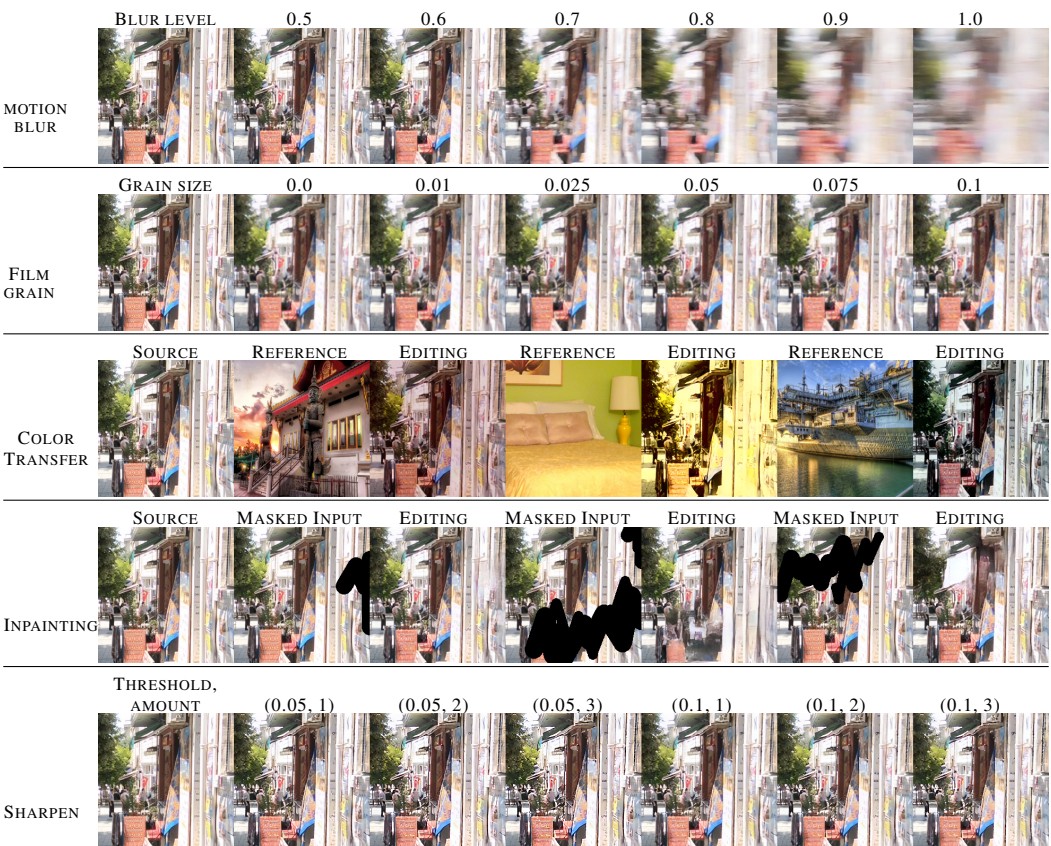

Figure 1: **Fine-grained user control on single-task editing.** Editing results were attained by progressively increasing the intensity level of the target effect (motion blur, film, grain, and sharpening), applying various arbitrary inpainting mask regions (inpainting), or randomly switching the reference image (color transfer).

## C  ADDITIONAL RESULTS

**Fine-grained user editing control.**    We demonstrate fine-grained user control on single-task editing in Fig. 1 and on multi-task editing in Fig. 2. Note that the user can incrementally increase the blur, film grain, and sharpening level. In addition, the user can transfer global color palettes from a variety of reference images and freely select different inpainting masks, spanning both small and large image regions. Fig. 2 shows that our model can handle multitask edits for motion blur, film grain, color transfer and sharpening but fails on inpainting task.

**Generalization to unseen datasets.**    To show the generalization capabilities of our approach to unseen data, we show additional qualitative comparisons on FFHQ dataset Karras et al. (2019) in Fig. 3. While our proposed method achieves comparable results to Palette++ and our task-specific

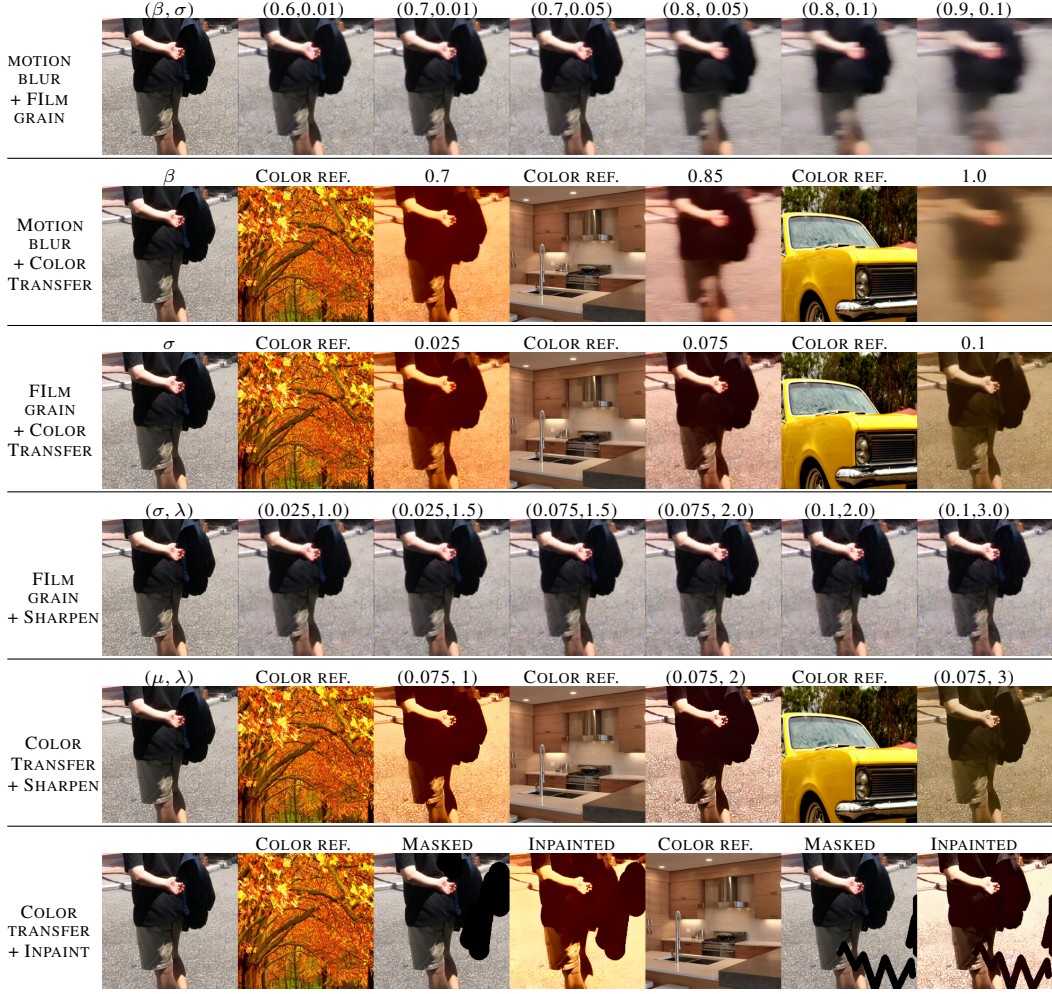

Figure 2: **Fine-grained user control on multi-task editing.** Editing results were attained by progressively increasing the intensity level of the target effect (motion blur, film grain, and sharpening), applying various arbitrary inpainting mask regions (inpainting), or randomly switching the reference image (color transfer).

baseline on single-task editing, it outperforms all baselines on multi-task editing. For instance, our approach faithfully reproduces color transfer effects even when graining and sharpening effects are additionally added to the input image. It also better preserves the blur and grain intensity level when both blur and grain are combined together.

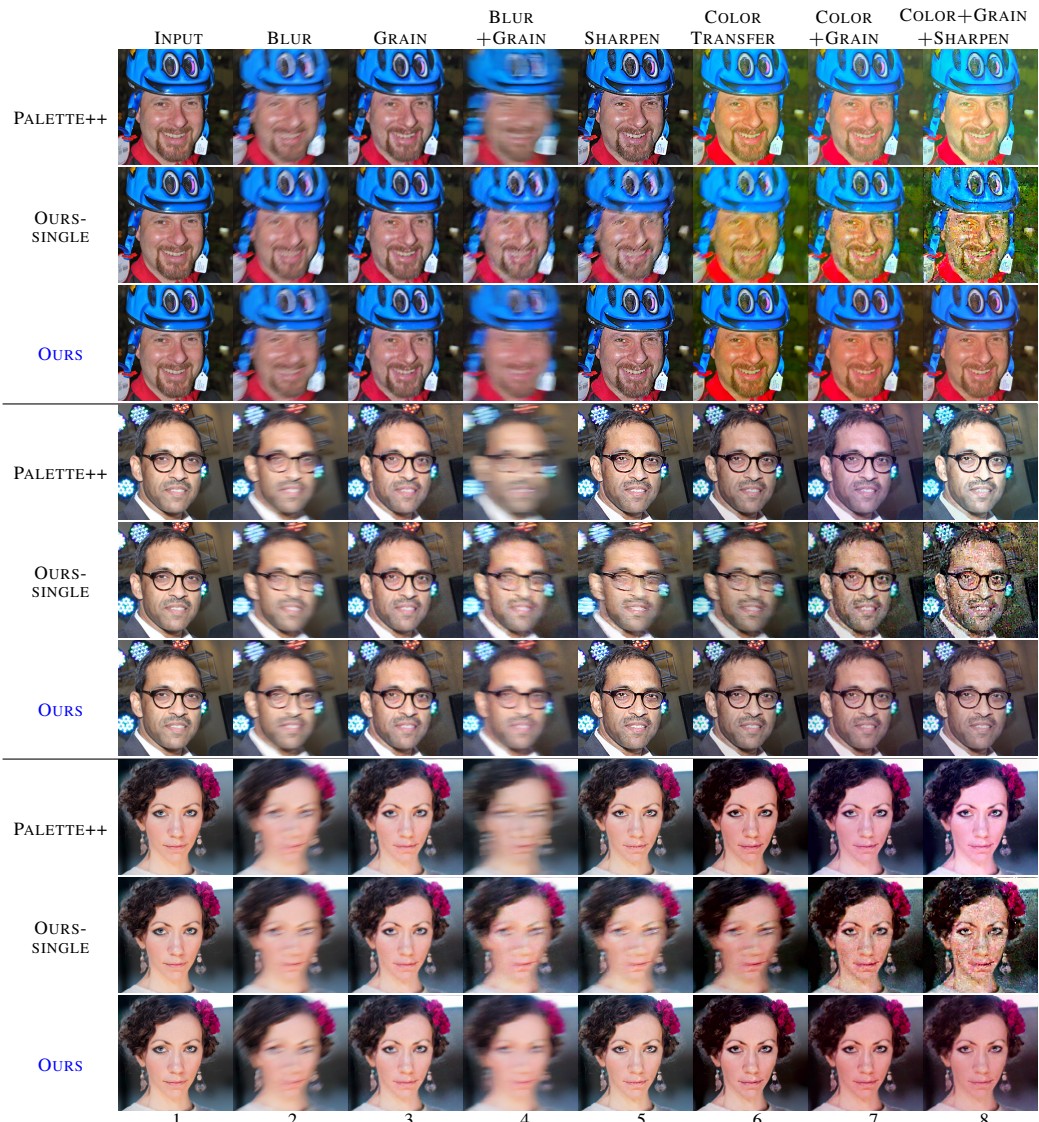

Figure 3: **Multi-task image editing results on FFHQ dataset.** We show additional single- and multi-task editing results on FFHQ Karras et al. (2019) using the model trained on our custom dataset.

## D    MULTI-TASK EDITING DISENTANGLEMENT

**Contrastive-based feature disentanglement.**    We provide further analysis of the experiments with the N-pair loss. To achieve better disentanglement for multitask editing, we added an additional N-pair loss to the training objective in our experiments. We have tested two configurations by applying the contrastive N-pair loss on 1) the bottleneck features of the denoising U-Net and 2) the label features $f_\theta(\mathbf{C})$. Figure 4 shows the t-SNE Van der Maaten & Hinton (2008) plots of the bottleneck features (left) and the label features (right) for our approach and the two additional experiments. The left column shows that both configurations with N-pair losses achieve better disentanglement of the bottleneck features than our approach. The right column shows that our approach already naturally disentangles the label features by design. However, the quantitative metrics in Table 5 of the main paper show that applying N-pair loss shows no noticeable improvements to the baseline on single-task edits, and even a drop in performance on multi-task edits. While the t-SNE plots show that the contrastive N-pair loss is trained successfully for latent feature disentanglement, it has in practice very little impact on the quality of the editing results. Applying contrastive constraints to diffusion models thus remains an interesting future research direction.

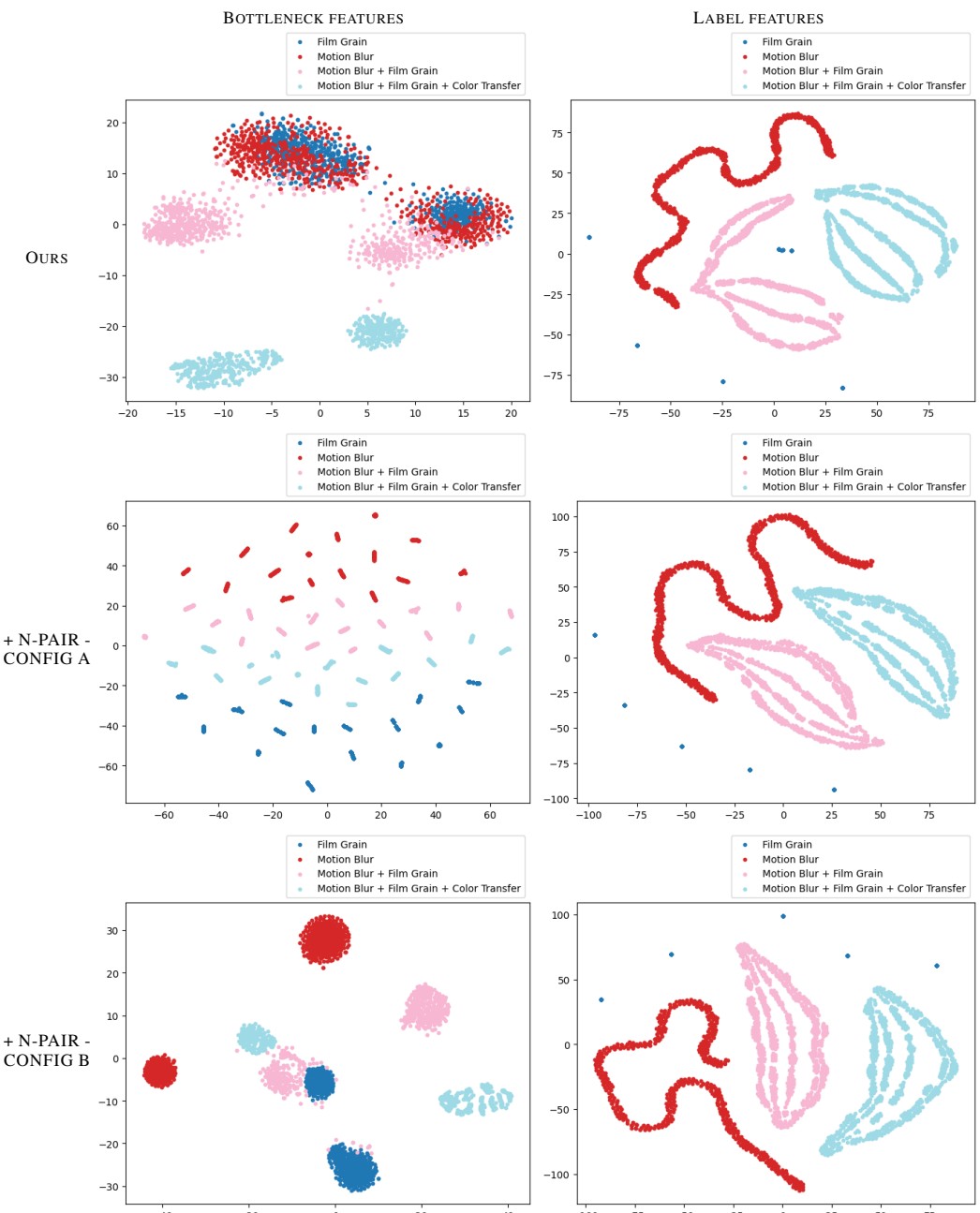

Figure 4: **Feature visualization.** *Left:* t-SNE plots of bottleneck feature maps. *Right:* t-SNE plots of input label feature maps. + N-pair Config A and B refer to our model trained with an additional contrastive N-pair loss applied to the bottleneck features and label features, respectively.

# E  LIMITATIONS

**Image quality and DDIM inversion.** Fig. 5 presents qualitative results obtained with and without DDIM inversion (the latter our choice). For each task shown on the top row, the left column shows the ground truth edits, while the right column shows the result obtained by our approach, with or without DDIM inversion applied during the sampling process. Note that the first column is the input sharp image. For each editing result, we also calculate $\ell_1$, SSIM, and LPIPS with regard to the ground truth edit. Although the results generated with DDIM inversion might look crisper, e.g., on the first image row, the edits do not faithfully reproduce the desired target effect, thus resulting in

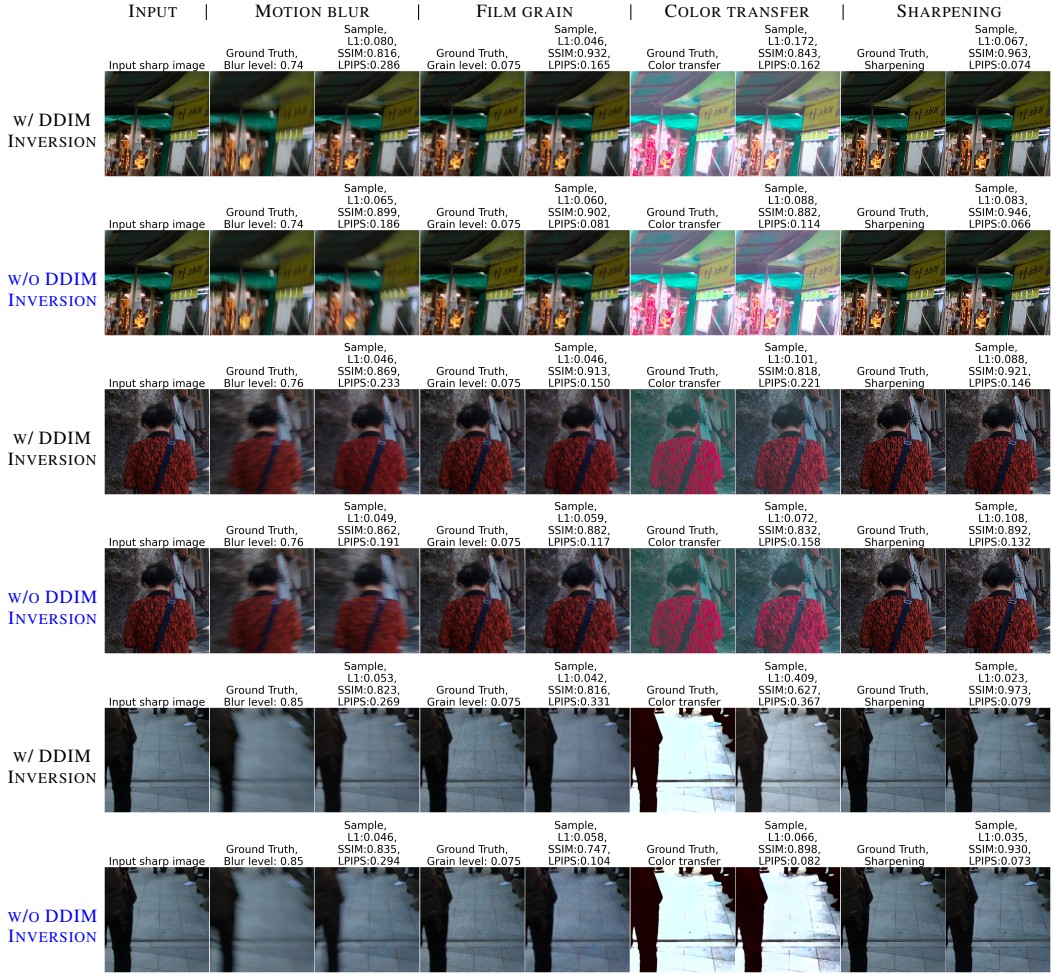

Figure 5: **Comparison between our approach and DDIM Inversion Dhariwal & Nichol (2021) on single-task image editing.** Blue denotes our proposed model.

higher errors as reported by the aforementioned image metrics. Thus, results produced without DDIM inversion attain higher-quality photo filter effects, yet at the expense of less crisp output images.