# OpenReview forum: "Multitask Image-to-Image Diffusion Models with Fine-Grained Control"
_ICLR.cc/2024/Conference — ICLR 2024 Conference Withdrawn Submission_

### Official Review · Reviewer_9BYu · 2023-10-28

**Soundness:** 2 fair
**Presentation:** 3 good
**Contribution:** 3 good
**Rating:** 5
**Confidence:** 4

**Summary:**

The authors propose a novel image-to-image diffusion model that is trained for multiple editing tasks. Thanks to its new conditioning framework, it can process compositions of multiple tasks, even though its training dataset only contains ground-truth image pairs for each individual task. The experimental results show that the proposed model performs well compared with baseline methods such as pix2pix and Palette.

**Strengths:**

- The proposed framework for conditioning is simple and should be easy to implement.

- The experimental results are somewhat amazing. Even though the model is trained to process each individual task, it is also capable of processing compositional tasks. It is also interesting to see providing binary task labels boost the performance in such tasks.

- Overall, the manuscript is well-written and easy to follow.

**Weaknesses:**

<Major ones>

- The novelty in methodology is somewhat marginal. The use of task labels has been becoming popular nowadays in the era of Transformer (e.g. [R1]), and using cross attention layers for conditioning would be one of the standard choices (e.g. [R2]).
  - [R1] "UniT: Multimodal Multitask Learning with a Unified Transformer," ICCV 2021.
  - [R2] "High-resolution image synthesis with latent diffusion models," CVPR 2022.

- Although the emerging ability for compositional editing tasks is interesting, it is not clear to see why or how it is obtained in the proposed method.

- The validity of the choice of evaluation metrics in the experiments is not clear. The perceptual metrics are commonly used to measure perceptual similarity between clean and degraded image. However, in this study, it is used to compute the similarity between degraded images with motion blur or file grain. Is it really suitable to evaluate whether the images are similarly degraded or not especially in the case of compositional editing tasks?

- It sounds unreasonable that the conditioning parameter of motion blur only is just a single scalar indicating a blur degree, because it is often represented by a blur kernel in the literature. In addition, how to obtain ground-truth values in the training dataset is not clearly described.


<Minor ones>

- I am not sure if "fine-grained control" is a proper term here. If we say "fine-grained" in the field of computer vision, we often expect sub-category level of semantics or pixel level control of manipulation. The controllability in the proposed method seems standard level, not fine-grained.

- The experiments with N-pair loss could be moved to the supplementary materials. Instead, it would be helpful for understanding disentanglelemnt obtained in the proposed method to provide some analysis on the output of f_theta as shown in top-right of Figure 4 in the supplementary materials.

**Questions:**

Please see weaknesses.

---

### Official Review · Reviewer_iY2j · 2023-10-30

**Soundness:** 1 poor
**Presentation:** 2 fair
**Contribution:** 2 fair
**Rating:** 3
**Confidence:** 4

**Summary:**

This paper presents a multi-task image-to-image diffusion model, trying to use a single model to fine-grained control image editing among multiple tasks, i.e., motion blur synthesis, film grain synthesis, color transfer, inpainting, and sharpening. Some experiments are coonducted to demonstrate the effectiveness of the proposed method.

**Strengths:**

1. The proposed method has the ability to use one model to deal with several situations.

**Weaknesses:**

1. The experimental comparison of methods is weak, too many image-to-image translation methods can be compared, yet, the authors only compared one  GAN-based method (Pix2pix, 2017) and one diffusion-based method (Palette, 2022).
2. The visual effect of the results does not look so remarkable, e.g., in fig.2, I can see the obvious difference between your results and the input (line INPAINT and line SHARPEN).

**Questions:**

1. The authors mention that there is less literature related to motion blur synthesis. Have the authors ever researched in the deblurring literature, there have been many successful techniques to synthesize motion blur, e.g., DeblurGAN [1], RSBlur [2].
2. Why did the author start with the inverse image restoration (motion blurring) instead of the image restoration task (deblurring)? Blurring an image seems very easy, a simple convolution with a blur kernel could achieve similar film-blurring effect.
3. How about the model inference time, model size and GPU usage？

[1] Blind Motion Deblurring Using Conditional Adversarial Networks, CVPR, 2018.
[2] Realistic Blur Synthesis for Learning Image Deblurring, ECCV, 2022.

---

### Official Review · Reviewer_Nn9D · 2023-11-01

**Soundness:** 3 good
**Presentation:** 3 good
**Contribution:** 2 fair
**Rating:** 6
**Confidence:** 3

**Summary:**

The authors design a multi-task image-to-image diffusion model in this paper, which equips diffusion models with the ability of compositing multi-task editing capabilities and also with more fine-grained control from users. Previous multi-task diffusion models like Palette only explores multi-task pixel-to-pixel mapping in a limited context. In comparison, this paper proposes a new input conditioning formulation that allows more flexible and fine-grained control over each task. For better evaluation on the proposed method, the authors also create a customized dataset based on the existing dataset REDS. Experimental results show that the proposed method achieves state-of-the-art performance on both single-task and multi-task image editing tasks.

**Strengths:**

++ The presentation of this paper is generally good and the method part is easy to understand.

++ The proposed method is of great practical use in the fields like film production, for example, adding motion blur and film grain on the captured pictures or videos.

++ The authors propose a dataset that might be useful for future researches of multi-task image editing.

++ The proposed method outperforms the previous multi-task diffusion model Palette and achieves the state-of-the-art performance in single-task / multi-task image editing.

**Weaknesses:**

-- I think this overall logic of this paper is good. The reason why my current rating is a little conservative is because I am not sure how much insight other people could draw from this paper, and how influential this paper would be in the related or broader fields. The reason for my concern is that I think the method of this paper can be viewed as a simple gating strategy for multiple tasks, where the proposed "binary task activation labels" serve as indicators for hard gating. Here I did NOT mean that it is not good to be simple and methods have to be complicated. In fact it is really valuable for a method to be "simple and effective". However, what I do feel concerned about is that there might not be adequate insight in this model design such that other researchers could not get much inspiration from this paper, making the overall contribution of this paper limited.

-- When creating your dataset, the ground truth of the edited images are obtained by off-the-shelf algorithms. I am wondering if we regard them as ground truth, then under this logic, the best edited results we could ever reach would be the results obtained by those off-the-shelf algorithms. Then, there will be no need to use the proposed algorithm, because its performance upper bound would be those off-the-shelf algorithms. Thus, I am not sure whether those ground truth generated by off-the-shelf algorithms would make the evaluation less reliable. I might be wrong of the above understanding, so please correct me if I am thinking wrong.

**Questions:**

-- During training the multi-task diffusion models, the authors mention that the label vector is always a one-hot vector, which means a random type of editing (or no editing at all) is applied on the image. I am wondering whether the label vector does not necessarily need to be a one-hot vector. In contrast, it could be something like a vector containing two 1's and the rest being 0's, indicating the image is simultaneously going through two types of editing. I think this is reasonable because the typical procedure of multi-task learning is learning multiple tasks together and simultaneously. In this case, during inference we no longer need to make the edits sequentially, instead we could perform all editing at once. I think an ablation of this implementation choice may be needed.

---

### Official Review · Reviewer_GVE7 · 2023-11-03

**Soundness:** 3 good
**Presentation:** 2 fair
**Contribution:** 2 fair
**Rating:** 5
**Confidence:** 4

**Summary:**

This paper addresses the challenge of multi-task image editing using a single model capable of multiple image-to-image translation tasks. The paper introduces a multi-task image-to-image diffusion-based framework. This is achieved through task conditioning by user defined input, which includes image conditions and task parameters for image editing applied via cross-attention. The model is trained on various editing tasks like motion blur, film grain, sharpening, and colorization. The proposed approach provides explicit user control over each task and outperforms existing methods when multiple editing is applied. The paper introduces a new benchmark based on the existing video dataset REDS.

**Strengths:**

- Introduces a new conditioning method for a diffusion model with explicit user control via task-specific scaling factors and binary task labels. This enables multiple types of image editing with a single model.
- The proposed method outperforms baseline methods on multi-task editing.
- Introduces a new benchmark dataset for image editing tasks.
- Multitask image editing is an underexplored research topic.

**Weaknesses:**

- The proposed model struggles with complete disentanglement of tasks in multitask editing setup. Adding one type of image edit affects the other editing properties. This is addressed as a limitation in the paper. The authors investigate contrastive learning to mitigate task entanglement which is presented in the supplementary material. However, contrastive learning did not resolve the issue.

- Single-task image editing performance is on par with baseline methods.

- The out-of-distribution test is only shown with qualitative examples without quantitative numbers. It is hard to claim that the proposed method generalizes to unseen images with few test examples.

- Some visual examples make it hard to tell the image editing effect. Figure 2 painting example does not have a masked region. It is hard to tell the difference between the input and ground truth of sharpen example in Figure 2.

**Questions:**

How does the method generalize to unseen images? More diverse test examples such as indoor scenes, scenic nature views, or aerial images would be more suitable for generalization tests.

Why only visual examples are shown for the generalization evaluation? Quantitative performance measures would be more appropriate to compare performance.

Single task performance of the proposed method is on par with the baselines (Table 2) while the proposed method outperforms the baselines on multitask setting (Table 3). I am concerned whether this performance gain is due to overfitting the training data, at the cost of generalization performance. If this is true, the applicability of the proposed method is limited, because off-the-shelf image editing methods used to generate benchmark dataset introduced in the paper is domain agnostic. I recommend adding more generalization experiments and analysis.